# Stem Growth of Horse Chestnut (*Aesculus hippocastanum* L.) under a Warming Climate—Tree Age Matters

**Roman Plichta \*** , **Luboš Úradníček and Roman Gebauer**

Department of Forest Botany, Dendrology and Geobiocoenology, Mendel University in Brno, Zemědělská 1, 613 00 Brno, Czech Republic
\* Correspondence: roman.plichta@mendelu.cz

**Abstract:** This research provides new information about the effect of drought on horse chestnut growth (*Aesculus hippocastanum* L.) in different ages. Global climatic scenarios predict a higher frequency of heatwaves and drought periods; however, investigations into the growth reaction of horse chestnut to drought are completely lacking. Approximately 50-year-old solitary, 100-year-old solitary, and 100-year-old canopy horse chestnut trees in a floodplain area were investigated. Growth reactions measured using automated dendrometers with respect to meteorological variables and water table depth were investigated during the years 2019–2021. Cambial activity was shown to be driven by tree age, as younger trees had higher stem radial increment rates. Both mature tree groups suffered from a low depth of water level and from higher sensitivity to meteorological variables, as growth was limited when mean daily vapor pressure deficit (VPD) exceeded 600 Pa. Together with a lower probability of growing days and a shorter growing season (GS) with earlier cessation of growth resulted in a lower total year radial increment (GRO) and basal area increment (BAI) when compared to younger trees. The young trees also exhibited lower tree-water-deficit-induced stem shrinkage (TWD) across all the studied years. Overall, horse chestnut trees in this floodplain area could be endangered by the decreasing level of soil water, with a greater age exacerbating the effects of drought. The year water deficit exceeded −340 mm in this locality every year, which has to be compensated for by regular flooding.

**Keywords:** dendrometers; drought; ecophysiology; flooding; phenology; senescence

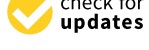



## 1. Introduction

Horse chestnut (*Aesculus hippocastanum* L.) is a deciduous tree species first introduced to western Europe during the 16th century, documented by the first printed illustration of the horse chestnut by Italian physician Pietro Andrea Mattioli in Prague in 1563 [1]. Although the native distribution is restricted to the Balkan peninsula [1], it rapidly became a very popular tree, which was planted as an ornamental or memorial tree in botanical gardens, parks, or avenues [2]. Horse chestnut was also planted more often along roads after Maria Theresa's urban forestry decree, which ordered the planting of trees along all new roads in order to orient travelers in fog and snow, increase wood production, and enhance the appearance of the landscape [3]. The large seed of the horse chestnut is also a valuable source of carbohydrates; thus, the tree was planted in forests in order to increase and diversify food resources. However, the presence of various metabolites makes the horse chestnut a less attractive food source for game [4].

One of the land areas which possess horse chestnut trees from different historical eras, planted as either ornamental trees, memorial trees, or as a source of food, is the "Soutok" floodplain forest area on the border of three states (the Czech Republic, Austria and Slovakia) [5]. This area was utilized for hunting by the Liechtenstein nobility from the 15th century onwards. As a result of the introduction of historical tree species and the utilization of this area as a game reserve up to the present day, individuals of horse chestnut

in the decay, senescent, mature, or juvenile phase can be found, forming a forest admixture of 0.08%. However, as the increasing timber utilization and field management in this area was in conflict with the regular inundation of floodplain forests, the area was recultivated, and flooding began to be artificially regulated from the 1970s [6]. This regulation of flooding has resulted in a significant decline in the ground water table [5,6], which may have an adverse effect on tree vitality.

Despite the importance of horse chestnut heritage and its aesthetic function, very little is known about its ecophysiology—specifically, its abiotic stress response. Most studies focus on the altered physiology of horse chestnut leaves after an attack by the invasive horse chestnut leaf miner (*Cameraria ohridella* Deschka & Dimic) [7,8]. The larvae of the moth mining within the leaf significantly shorten the leaf lifespan, leading to defoliation well before normal leaf-fall in the autumn [8]. Together with a reduction in the photosynthetic leaf area during mining, the infested trees exhibit lower annual net primary productivity by dozens of percent compared to uninfested trees [9,10]. Nevertheless, premature leaf shedding does not result in significant total energy loss and horse chestnut trees do not face a serious risk of dieback in the following years [10]. However, the diminished primary productivity has negative long-term consequences for the storage of carbohydrates in roots and branches, resulting in a reduction in seed biomass, growth, and tree vitality [9]. Moreover, depleted starch reserves could also have negative consequences for the drought resistance and resilience of horse chestnut, as carbohydrate content is vital for surviving long-term and re-occurring drought conditions, when lower stomatal conductance minimizes water loss and limits photosynthesis at the same time [11,12].

Another important factor determining tree response to drought is tree age [13,14]. Despite the fact that trees have no programmed senescence [15], the tree tends to grow slowly with increasing age and size and is more likely to die [16]. However, the situation is more complex. Since tree mortality during drought is probably the result of the loss of xylem hydraulic function [13], xylem embolism resistance presumably does not differ with age [14]. Old trees, bigger in size, have a larger leaf area exposed to extreme atmospheric conditions during drought periods, increasing water consumption by the individual tree. On the other hand, old trees can benefit from more complex root systems reaching deeper soil water reserves. The higher mortality with tree age thus may not be a result of hydraulic failure per se, but could be accompanied by other factors predisposing the tree to die, such as the accumulation of damage from pathogens and reactive oxygen, somatic mutations, or simply size-to-growth limitations [15]. Regardless of the age of the tree, we know very little about the effect of drought on the growth and vitality of horse chestnut.

To shed more light onto horse chestnut drought reactions, the stem growth of two tree age classes (young and mature) was monitored for three consecutive growing seasons. Artificially managed floodplain forests in the Soutok region, those in the warmest climate of the Czech Republic, were selected, as these conditions exacerbate the deterioration caused by severe drought conditions. We hypothesized that, regardless of tree age, the altered soil water conditions during drought would affect all age classes, though mature trees would respond more sensitively as a result of ageing.

## 2. Methods

### 2.1. Study Site and Experimental Design

The study site, managed as a game reserve, is located in the Soutok floodplain area, South Moravia, Czech Republic (48°43′43″ N, 16°53′42″ E), at an altitude of 200 m a.s.l. Long-term (1981–2010) average annual air temperature and precipitation are 10.6 °C and 587 mm, respectively [17] (Figure S1). The predominant soil type is Chernozem. Spring inundations are artificially managed, and their durations differ according to the water availability in the Thaya River. Inundations occurred on 4–9 March 2019, 3–5 March 2020, and 14–22 May in 2021. The water table is approx. 3 to 4 m below the soil surface, situated in the sandy soil layer. Five horse chestnut (*Aesculus hippocastanum* L.) trees in the young reproductive age class (approx. 50 years old) and 10 trees in the mature reproductive age

class (approx. 100 years old; transition between mature and old reproductive class [18]) were chosen to evaluate growth reactions to changing meteorological variables during the growing seasons 2019–2021. All young trees and five mature trees were solitary trees with no suppressed crowns, sparsely distributed on a meadow with the shortest distance between each other higher than the tree height, while another five dominant mature trees were chosen from the open canopy with no admixture of other tree species. This resulted in the observation of three tree groups: young solitary (YOUNG$_S$), mature solitary (MATURE$_S$), and mature canopy (MATURE$_C$). The young canopy trees were not present in our research area; thus, this group was omitted. Tree groups were selected to be as close as possible, to minimize the effect of the different site conditions, which are in this alluvial site minimal. Only representative healthy trees with the same social status and similar dimensions were selected within the studied groups. The average tree height and circumference, respectively, were $9 \pm 3.3$ m and $58.8 \pm 16.1$ cm for YOUNG$_S$, $15.4 \pm 2.5$ m and $311.9 \pm 87.8$ cm for MATURE$_S$, and $21 \pm 1.2$ m and $146 \pm 24.6$ cm for MATURE$_C$ trees.

### 2.2. Meteorological and Soil Water Variables

Air temperature (T), relative air humidity (RH), and wind speed (WS; all at 2 m height) as well as global radiation (GR) and precipitation (Prec) were measured every 15 min using an automatic meteorological station (EMS Brno, Brno, Czech Republic) on the open meadow approx. at the center of the study site. The reference evapotranspiration (ET$_0$) was calculated [19,20], where net radiation was estimated as 70% of the GR. The monthly water balance was calculated as the difference between monthly Prec and ET$_0$ and the water balance below zero was taken to be the water deficit. It is worth noting that this is a very rough characterization of water balance, as evapotranspiration from grass without any stomatal limitation and under fully watered conditions was assumed. To provide soil water characteristics, the monthly average soil water table depth (Depth) was taken from the near shallow borehole covering the normal period between 1980 and 2011 and the studied period 2019–2021 [17]. Moreover, the monthly standardized precipitation-evapotranspiration index (SPEI) was calculated on the basis of local data from the meteorological station and regional normal data from the Global SPEI database [21] to a 1-degree spatial resolution (48°45′ N, 16°45′ E) and using R [22] and the SPEI package [23]. On the basis of the SPEI, drought severity can be scaled as no-drought (SPEI > −0.5), mild drought (SPEI −0.5 to −1), moderate drought (SPEI −1 to −1.5), severe drought (SPEI −1.5 to −2), and extreme drought (SPEI < −2) [24].

### 2.3. Stem Radius Variation

Stem radius variation was measured every 15 min in all trees using automatic point dendrometers (Tomst, Prague, Czech Republic) installed at a height of approx. 2 m. Dendrometers were installed on the north-facing side of the stem. To account for possible variations around the larger stem circumference of the mature individuals, these trees were equipped with an additional dendrometer on the east-facing side and the data from both sides were averaged. The rhytidome was removed at the place of the installation. The stem diameter at a height of 1.3 m was measured using a diameter tape, and the tree height was measured using a TruPulse 200L laser rangefinder (Laser Technology, Centennial, CO, USA). To illustrate the severity of horse chestnut leaf miner (*Cameraria ohridella* Deschka & Dimic) attack, three leaves from the middle canopy of each tree were sampled on 4 August 2021 and scanned, and the percentage of leaf mines to total leaf area was assessed using ImageJ [25].

### 2.4. Growth Reaction Assessment

Dendrometer time series were checked and cleaned (from the effects of sub-zero temperatures, re-installation jumps, and errors) in R using the interactive authors-compiled R package PLOTeR (Plichta, in preparation). The zero-growth concept [26] was utilized to partition the raw dendrometer data into growth-induced irreversible expansion (GRO) and

tree water-deficit-induced stem shrinkage (TWD). TWD is proven as a good measure of tree water status, while TWD is accepted to be a good predictor of midday xylem water potential [27]. Since the shrinkage in bark cells (including phloem) is the predominant driver of TWD and the thickness of bark (after the removal of rhytidome) is taken as constant across observed ontogeny stages, TWD data were not normalized to stem diameter. In contrast, to account for the effect of different tree dimensions on GRO, these were normalized to basal area increment (BAI):

$$BAI = \pi(r + GRO)^2 - \pi r^2 \tag{1}$$

where $r$ is the radius of a given tree at the beginning of the experiment.

The growing season (GS) start, end, and length were determined from a mean 14-day moving average of GRO rates, which have a typical bell-shaped progress during the GS, after removal of the lowest 5% of GRO rates. For testing the effect of SPEI and water deficit on stem radial changes, GRO and TWD were standardized to minimize the effect of the growing season. Monthly GRO rates and the monthly sum of TWD were calculated and expressed as relative to maximum and minimum values, respectively, within each month and tree group.

*2.5. Statistical Analysis*

Generalized linear models (GLM) were used for linear regressions and GLM followed by Tukey's honestly significant difference (HSD) test were used to assess the difference in phenology (GS start, end, and length) between tree groups and between the studied years. Linear mixed-effect models (LME) with the random factor of years nested in individual trees followed by HSD test were used to determine the difference between the yearly BAI, GRO and TWD. In order to assess the explanatory power of the environmental variables on tree growth, tree group-specific generalized linear mixed effect model (GLMM) with GRO and TWD as response variables (link = "log"), with scaled fixed effects and year nested in tree as random effects were calculated based on the monthly averaged or summarized values. Because of the collinearity of several meteorological variables, the only GR, T, VPD, Prec and Depth were considered as fixed effects. The only months covering the GS were included in the analysis. In order to visualize the effect of VPD on tree growth rates on a daily basis, the general additive models (GAM) were fitted between the maximum GRO and BAI rates for each 100 Pa bin of mean daily VPD. R was used for statistical analysis using the "glm" function from the stats package, the "lme" function from the package nlme, the "glmer" unction from lme4 package, while Tukey's test was performed using the "glht" function from the multcomp package. The GAM models were performed using the "gam" function from the mgcv package.

## 3. Results

### 3.1. Meteorological Conditions

While average yearly temperature (T) and precipitation (Prec) were not so variable (Table 1), several drought stress periods were observed during the three studied years (Figure 1). Meteorological conditions in winter and early spring resulted in repeatedly low regional SPEI during February (2019 and 2021) and March (all studied years) (Figure 1). High reference evapotranspiration ($ET_0$) with lower Prec in June 2019 and 2021 led to the worst regional meteorological conditions with a minimal SPEI below $-1.5$ (Figure 1). This corresponded with the highest VPD (4.16 kPa) measured on 6 July 2019 and the highest temperature (33.2 °C) measured on 26 June 2019. Moreover, the bad meteorological conditions in 2019 were accompanied by the very low depth of the water table (Figure 1). Though the regional SPEI was more positive in 2020, in contrast to the other years, the higher evaporative demands in this year led to the highest cumulative water deficit of $-516$ mm (Figure 1). Nevertheless, the high evaporative demands of the studied floodplain area resulted in a cumulative water deficit lower than $-340$ mm every year (Figure 1).

Although the regional SPEI and local water deficit were linearly related to each other ($p = 0.007$), this relation was weak ($R^2 = 0.17$).

**Table 1.** Average yearly temperature (T) and total yearly precipitation (Prec) on the studied locality.

| Year | T | Prec |
|------|------|------|
| | (°C) | (mm) |
| 2019 | 11.2 | 581 |
| 2020 | 10.3 | 580 |
| 2021 | 9.52 | 478 |
| 1981–2010 | 10.6 | 587 |

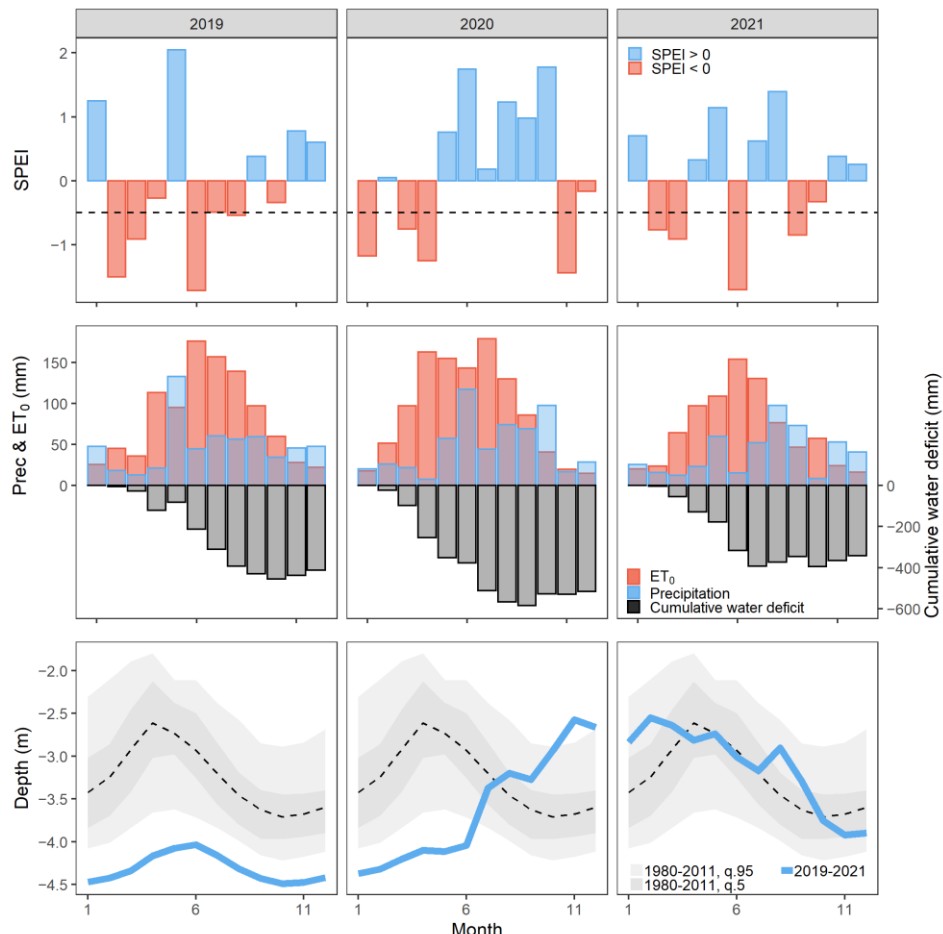

**Figure 1.** Monthly standardized precipitation-evapotranspiration index (SPEI; upper panels), monthly precipitation (Prec), reference evapotranspiration ($ET_0$), and cumulative water deficit (middle panels), and monthly mean water table depth (Depth; lower panels) during the years 2019–2021. Spring and summer drought accompanied with low depth resulted in harsh environment conditions in 2019. See Figure S1 for long-term trends.

*3.2. Phenology*

On average, the growing season (GS) started at the end of April (DOY 119 ± 4) and finished at the beginning of August (DOY 214 ± 12) (Figure 2). The GS of individual trees began between mid-April and the end of May (DOY 106–143) and was completed between the end of June and mid-September (DOY 172–258). On average, all tree groups started their GS simultaneously, while YOUNG$_S$ trees finished their GS later ($p = 0.002$), which was reflected in their longer GS ($p = 0.004$) compared to both mature tree groups (Figures 2 and S2). Generally, the length of the GS was clearly defined by the end of the GS

across all tree groups ($p < 0.001$, $R^2 = 0.86$) (Figures 3 and S3), while no effect of the onset of growth was observed ($p = 0.2$, $R^2 = 0.02$). MATURE$_S$ trees were more affected by horse chestnut leaf miner than the MATURE$_C$ and YOUNG$_S$ tree groups on 4 October 2021, as their functional leaf area was 19 ± 16%, 78 ± 11% and 74 ± 17%, respectively.

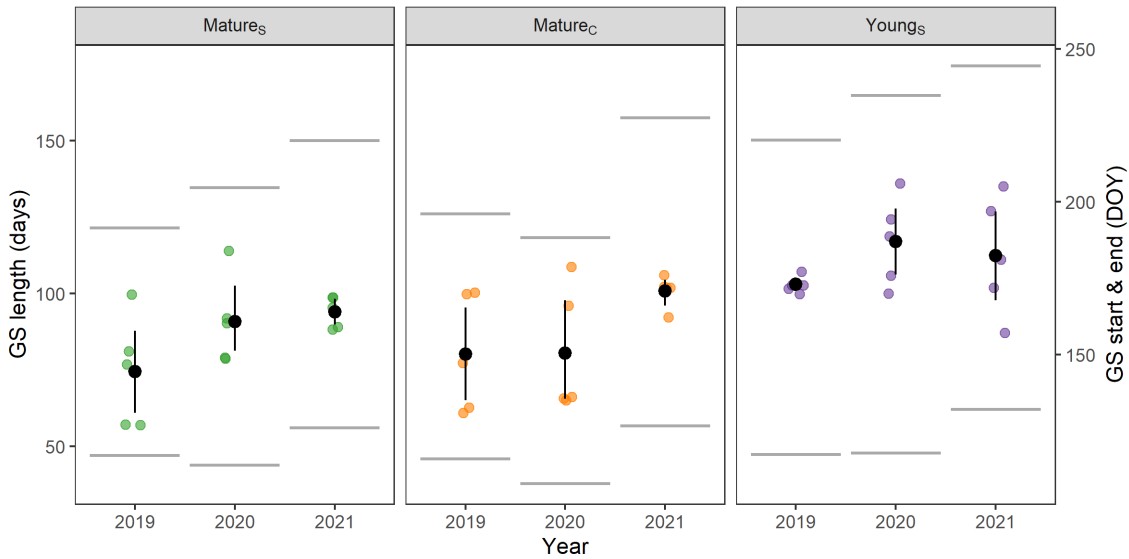

**Figure 2.** Mean growing season (GS) length (black points) with confidence intervals (black vertical error bars) and mean start and end of GS (lower and upper horizontal lines, respectively) for three studied tree groups during the years 2019–2021. Young solitary (YOUNG$_S$) trees finished their GS later and had a longer GS compared to both MATURE$_S$ and MATURE$_C$ groups. In 2021, the start and end of the GS was on average shifted to later days of the year (DOYs). See also Figure S2.

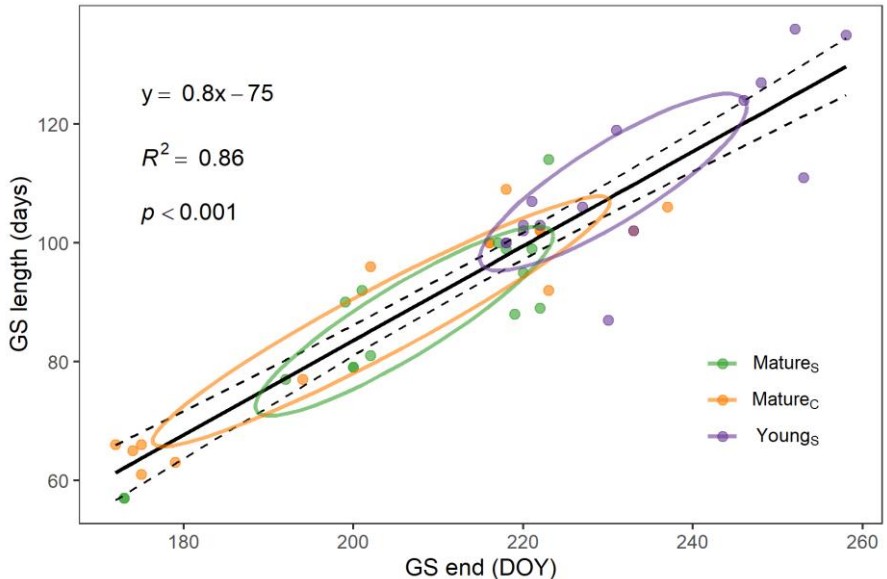

**Figure 3.** Linear relationship between length of growing season (GS) and day of year (DOY) when GS ended. Ellipses depict the t-distribution (confidence level 0.5) of data points from the geometric centre of each tree group and point to a longer GS which finished later in the YOUNG$_S$ tree group, while all tree groups showed a similar relationship (see also Figure S3).

### 3.3. Secondary Growth Dynamics

The yearly radial growth increment (GRO) of the YOUNG$_S$ group was more than five times higher than that of the other two mature groups in all studied years (Figure 4, Table 2). The YOUNG$_S$ group grew radially approx. 3 to 4 mm every year while both

mature groups grew radially approx. 0.6 mm every year. After translating GRO into the normalized basal area increment (BAI), the differences between groups diminished, but the general trend of YOUNG$_S$ trees exhibiting the highest total growth still remained (Figure 4, Table 2). The MATURE$_S$ and MATURE$_C$ groups had approx. 85% and 38% of total yearly BAI, respectively, when compared to the YOUNG$_S$ group. The lower total BAI of the mature groups was affected not only by the lower daily GRO rate, but also by the lower number of days with non-zero GRO (i.e., growing days) during the GS. Nevertheless, when comparing the maximum daily BAI rate, the MATURE$_S$ group had a three-times-higher maximum daily BAI rate than the MATURE$_C$ group and a two-times-higher maximum daily BAI rate than the YOUNG$_S$ group (Figure 4). The GRO daily rates of the tree groups differed especially in the summer, when an abrupt decrease in or complete cessation of secondary growth typically occurred in most of the MATURE$_S$ and MATURE$_C$ trees. This cessation of tree growth differed between years and occurred between the beginning of July (2019) and the middle of August (2021) (Figure 4).

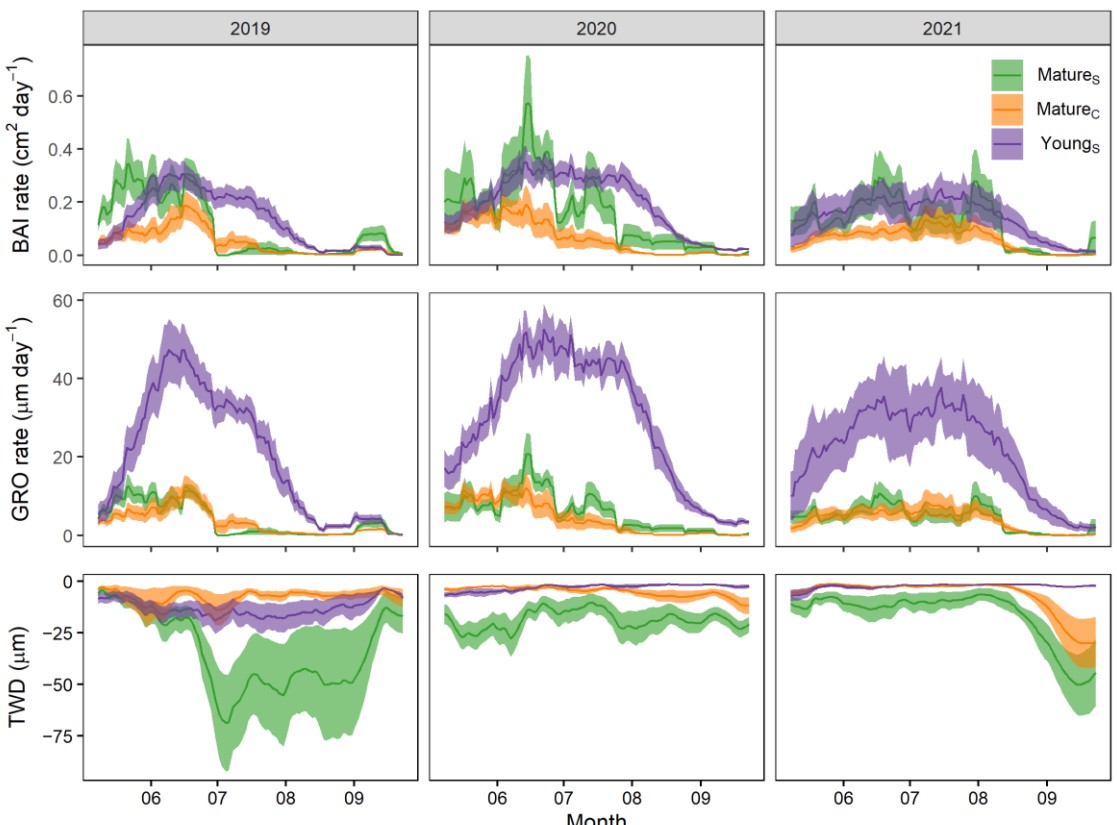

**Figure 4.** The average rate of the basal area increment (BAI), stem radius increment (GRO), and tree water deficit indicator (TWD) smoothed by a 14-day moving average and shown for the studied years 2019–2021. The abrupt cessations of the GRO rates of both mature tree groups are visible during the second half of the years. Colored area depicts standard errors.

The MATURE$_S$ group exhibited the lowest tree-water-deficit indicator (TWD; i.e., the highest drought stress), as it reached a minimum TWD of $-102 \pm 59$ µm, while the MATURE$_C$ and YOUNG$_S$ groups exhibited similar minimum TWD of $-50 \pm 25$ µm. The yearly sum of the TWD of the MATURE$_S$ group was also the lowest every year (significant across the years, $p = 0.01$) (Figure 4), as the MATURE$_C$ group reached on average 30% and the YOUNG$_S$ group only 13% of the yearly TWD of the MATURE$_S$ group.

**Table 2.** Estimates and confidence intervals (CI) derived from linear mixed effect models (LME) for total year basal area increment (BAI), stem radius increment (GRO) and tree water deficit indicator (TWD) of individual tree groups and the variance partitioning of the random effects (years and trees). Letters indicate significantly different tree groups. See Table S1 for details.

| | | MATURE$_C$ | MATURE$_S$ | YOUNG$_S$ | Tree | Year |
|---|---|---|---|---|---|---|
| | | Estimate [95 CI] | | | % | % |
| BAI | (cm$^2$) | 5.9 [3.3–10.6] [a] | 13.9 [7.8–25] [a,b] | 17.2 [9.6–31] [b] | 73 | 21 |
| GRO | (μm) | 408.8 [252.2–662.7] [a] | 531.5 [158.2–1786] [a] | 2668 [794–8964] [b] | 67 | 26 |
| TWD | (μm) | 488.4 [296.3–804.9] [a] | 1914 [546.6–6705] [b] | 184.7 [52.7–647] [c] | <1 | 96 |

*3.4. Environmental Factors and Tree Growth*

The generalized mixed effect models (GLMM) of monthly data showed that the GRO rate was positively related to the Prec, GR and the VPD and negatively to the T and Depth (Figure 5, Table S2). The effects of environmental variables followed the same pattern for GRO of all studied tree groups, but the size of effects was lower in YOUNG$_S$ trees. Contrary to the GRO, the opposite effect of the VPD and Depth was observed for TWD of MATURE$_S$ and YOUNG$_S$ groups, while MATURE$_C$ trees behaved differently. Overall, the Depth plays an important role in determining both the GRO and TWD in the studied floodplain area.

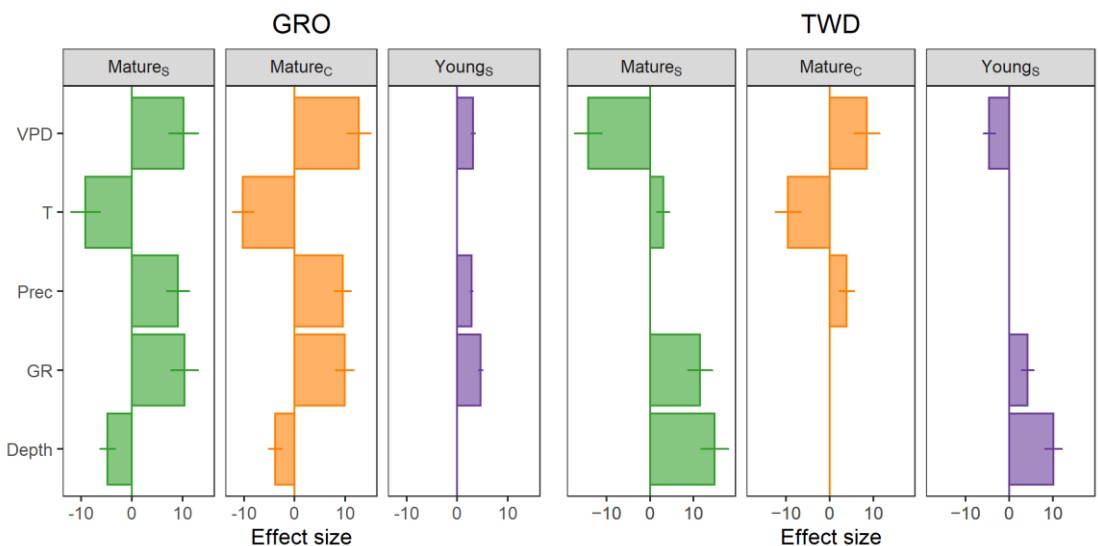

**Figure 5.** Effect sizes of environmental variables on monthly growth rate (GRO) and monthly tree water deficit indicator (TWD) derived from GLMM models (see Table S2). Depth—average monthly water table depth, GR—average monthly global radiation, Prec—monthly sum of precipitation, T—average monthly temperature, VPD—average monthly vapor pressure deficit.

When observing the effect of the VPD on maximal GRO rates on daily resolution, the opposite effect was observed contrary to the monthly data, where seasonality played a predominant role (Figures 5 and 6). The daily maximum GRO of each group negatively scaled with the VPD and it showed that YOUNG$_S$ trees had higher growth rates under higher VPD conditions (Figure 6). When the VPD was related to daily maximum BAI, the differences between groups observed for GRO under conditions of higher VPD disappeared (Figure 6) and, in contrast, the MATURE$_S$ group showed higher BAI under low VPD conditions compared to the other groups.

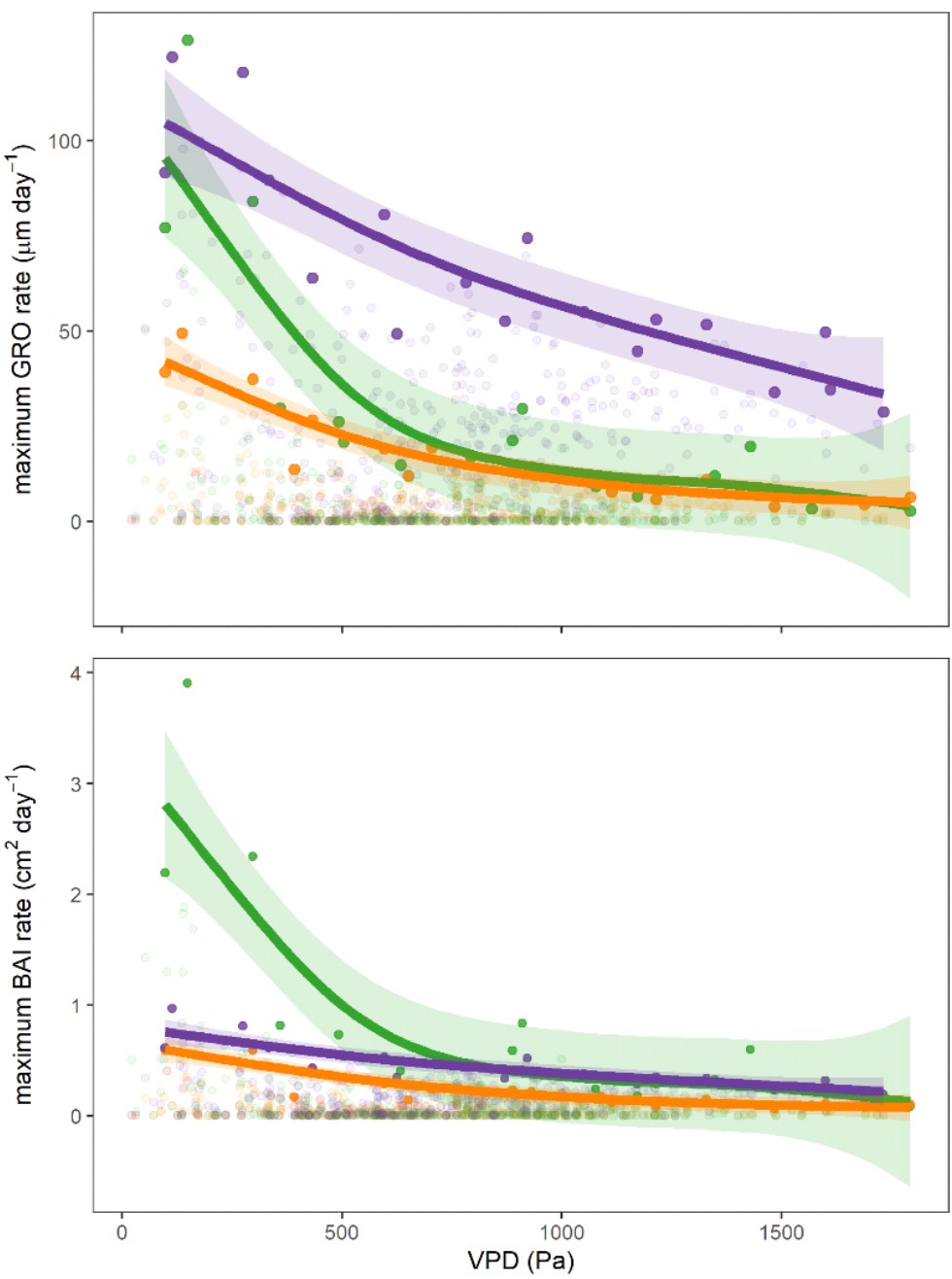

**Figure 6.** The relationship of mean daily VPD to maximum daily stem radius growth rate (GRO; upper panel) and maximum daily basal area increment rate (BAI; lower panel; larger solid points) smoothed by general additive models (GAM; thick solid lines). Colored area depicts confidence intervals of GAM models; smaller lighter points depict the relationship between GRO and BAI to VPD not fitted by GAM models (see Methods for more information). All GAM models were significant ($p < 0.001$) with $R^2 > 0.73$ using the formula y ~ s(x, bs = "cs") with the "REML" method.

## 4. Discussion

### 4.1. Meteorological Conditions

The studied floodplain locality is one of the warmest areas in the Czech Republic, with very low precipitation and high reference evapotranspiration ($ET_0$). The long-term trends for this region show slightly increasing total precipitation with lower precipitation intensity [28,29]. The total precipitation in this region in 2019 and 2020 nearly met the long-term average of 587 mm [17]; however, the total precipitation was 109 mm below the long-term average in 2021. On the other hand, an opposite long-term trend is observed

for the spring period, when decreasing precipitation and increasing temperatures are observed [28–30]. Correspondingly, a low amount of precipitation in the winter and early spring periods during the three studied years was also measured in our studied locality. All these factors led to a significant water deficit of around −400 mm every year. These environmental conditions have significant implications for flooding during the spring period, which is a necessary source of water for the floodplain forest ecosystem. In addition, in the studied area, flooding has been regulated since the 1970s and during very warm years with dry winter and spring periods the flooding is very short or does not occur at all. These conditions have led to substantial inter-annual variation in groundwater levels with the long-term trend of a declining water table [6], as could be seen in year 2019. All these factors have a strong negative impact on tree growth and vitality [6].

*4.2. Phenology*

In our study, significant differences between young and mature trees were observed in respect to length and the end of the growing season (GS), while no differences were observed in the onset of the GS. This is in contradiction to general findings, where young angiosperm trees have been observed to flush earlier than mature trees and start leaf senescence at the same time, e.g., [31–33]. For example, mature trees of Ohio buckeye (*Aesculus glabra* Willd.) showed bud burst 10 days after seedlings [32]. It is assumed that the earlier flushing of seedlings could be the result of higher temperatures near the soil surface where young trees grow [32,34]. The general later leaf flushing of mature trees could also be attributed to the longer distance between leaves, roots and stem, this resulting in a longer time to mobilize resources and increase phytohormones to levels inducing leaf flushing and secondary growth [35,36]. It should be noted that the phenology in this study was based on data from the secondary growth of the tree trunk and not from bud development. Nevertheless, it was observed that the secondary growth of ring-porous tree species was tightly coupled to bud burst and leaf flushing [37–39].

The onset of the GS varied not only between young and mature trees across the years, but also varied substantially among individual trees. This was the reason for the poor relationship between the start of the GS and the monthly or seasonal sum of meteorological variables. Although the start of the GS is largely determined by previous year conditions, especially drought [40–42], no significant trend was observed at the start of the GS in either young or mature trees after the intensive summer drought in 2019. The high variability in the start of the GS between years also indicates that temperature had a greater effect on the start of the GS than photoperiod, as also observed by others, e.g., [43–45]. Correspondingly, horse chestnut grown in Geneva, which has a warmer climate compared to our study, started the GS on average 80 days earlier [46].

In contrast to the onset of the GS, the timing of the GS's end was the predominant factor determining the length of the GS within and across the studied tree groups. The shorter GS with an earlier end in 2019 was presumably the result of the summer heatwave [47,48]. This means that current year environmental conditions had a strong impact on the length of the GS. Moreover, the GS generally tended to be prolonged across the studied years, though no significant relationship was found between the end of the GS and the monthly or yearly characteristics of meteorological conditions. It is obvious that tree phenology is driven by many factors, such as environmental conditions from the previous as well as the current year, soil conditions, genotype, tree age, and tree vitality, and thus large experiments with long-term observations would be needed to fully determine horse chestnut phenology [49]. In addition, besides meteorological conditions, secondary growth phenology could be affected by horse chestnut leaf miner larvae (*Cameraria ohridella* Deschka & Dimic), which feed on pre-senescent leaves. Therefore, the longer GS of younger trees in our study could be the result of the lower leaf deterioration at the end of the growing season compared to mature trees.

### 4.3. Secondary Growth and Water Stress

The secondary growth dynamics of the tree stem differed greatly between young and mature trees. In particular, young solitary trees had a five-times-higher yearly radial increment (GRO) compared to both mature groups. This corresponds to the general trend of decreasing GRO with age and tree size [50]. Indirectly, it also means that young trees exhibited higher cambial activity than mature trees [51], as radial increment is characterized by the cell size and number of cell layers produced by cambium [52]. It could be expected that the overall number of living cells increases with tree dimensions, as the periderm, phloem, cambial zone, and sap wood are strictly derived from organ circumference and tree height. However, tree size is limited biomechanically [53] and hydraulically [54], this in turn limiting the total leaf area [55,56] and having the strongest impact on growth rate [57]. As a consequence, older trees can decrease the activity of cambial cells to limit the number of living cell layers in the phloem, cambial, and sap wood zones [58,59]. On the other hand, the mature trees in our study exhibited a higher maximum basal area increment (BAI) daily rate, which indicates that despite their lower cambial activity, they managed to produce more biomass within a short time period than young trees [50]. Nevertheless, the young trees exhibited total yearly BAI 2.6 and 1.18 times higher than mature canopy and solitary trees, although their stem circumferences were two and four times smaller, which is related to the higher cambial activity of the young trees and their higher number of growing days (days with GRO > 0).

The higher probability of growth of young solitary trees corresponds to their sustained growth rate during days with higher vapor pressure deficit in the atmosphere. Vapor pressure deficit is one of the main drivers of stomatal regulation. Stomatal conductance decreases during days with higher VPD, when evaporative demands exceed the ability to transport water to the leaves [60–62]. However, this is in contradiction to the positive effect of VPD on monthly GRO rates observed by GLMM in our study. The positive effect of VPD on monthly scale arises from the higher GRO rates in the middle of the growing season, where the highest VPD conditions also occur. On the other hand, this relationship should be predominantly negative, when focused on a daily scale. In our study, mature trees significantly reduced stem growth when the mean daily VPD exceeded 600 Pa, while young trees grew even on days with a daily mean VPD above 1.500 Pa. In turn, the lower number of growing days in mature trees was coupled with the increasing occurrence of drought periods assessed by the tree water deficit indicator (TWD). In addition, absolute TWD values were higher in mature trees, signifying increasing water stress with the increasing age or size of the trees [63]. The periods with more pronounced TWD in summer were also accompanied by the cessation of GRO and the shorter GS of mature trees. These observations also correspond with the highest negative effect of decreasing water table depth (Depth) on GRO and positive on TWD of solitary mature trees, which were considerably larger than other tree groups. Moreover, summer water stress and growth cessation also coincided with horse chestnut leaf miner attack, as larvae gradually mine from May, with feeding culminating in June or July [64]. Nevertheless, the effect of an increasing percentage of mined leaves on the TWD dynamic is unknown and would be an interesting area for future research.

### 4.4. Environmental Factors and Secondary Growth

After the unprecedented drought of 2018, an even more pronounced drought spread across a large part of Europe in the summer of 2019 [47,48]. It corresponded with very low GRO rates in both young and mature trees in July and August 2019. Moreover, the meteorological drought conditions in 2019 were coupled with the abnormally low Depth. In 2019, the total GRO and BAI of young and mature trees were the lowest from the studied years, but the mature trees were more affected. Although mature trees can have larger root systems than young trees, they do not necessarily have better access to deeper soil layers, which are an important source of water during drought [65]. The reason for this is that the maximum rooting depth might not be different between 50-year-old and 100-year-old

trees, as soil properties [66] and soil hydrology, including the depth of the water table [67], could have a considerable impact on rooting depth. In addition, mature trees have a larger leaf area, which could disadvantage them during drought periods, as it increases the evaporative area.

The legacy effect of severe drought conditions on tree growth in the following years was observed in several studies, e.g., [68,69]. In spite of this effect, the horse chestnut trees in our study increased their growth by approx. 30% regardless of tree age in 2020 after the summer droughts of 2018 and 2019. However, the growth rates of the trees again decreased in 2021, though the TWD values were close to zero during much of the year, pointing to a possible legacy effect. Moreover, the lower GRO rates in 2021 compared to the previous year were not explained by water deficit nor the SPEI. For example, the positive SPEI during the GS in 2020 was coupled with higher growth, regardless of the significant cumulative water deficit at the site. This disconnection between growth characteristics and water deficit presumably arises from the soil water balance, as the soil water reserves started to be replenished in 2020, which was indicated by increasing Depth. Soil water content and soil water level dynamics have been found to be better indicators of tree growth than atmospheric conditions [11], and the large effect of Depth on TWD and GRO in our study is in accordance with these observations [70].

## 5. Conclusions

The age of a horse chestnut tree plays a dominant role in its growth sensitivity to meteorological and hydrological conditions, where the dryness of the atmosphere coupled with the depth of the soil water level were determined as the main factors affecting tree growth and tree water deficit during the growing season. The year water deficit exceeded $-340$ mm in the studied locality every year, which has to be compensated for by regular flooding. Horse chestnut trees in this floodplain area could be endangered by the decreasing level of soil water, with greater age exacerbating the effect of drought. Moreover, the suggested coincidence of the horse chestnut leaf miner attack with tree growth cessation and tree shrinkage would be an interesting area for future research.

**Supplementary Materials:** The following supporting information can be downloaded at: https://www.mdpi.com/article/10.3390/f13101677/s1, Figure S1: The average monthly mean temperature (red color) and monthly precipitation totals (blue color) for the period 1980–2011. Dashed line shows average, darker colored area is 50% quantile and lighter colored area is 95% quantile. Thick solid lines prescribe the temperature and precipitation during the studied period 2019–2021; Figure S2: Boxplots with median and interquartile range for average growing season (GS) length, start and end of studied tree groups. Different letters show statistical difference among tree groups assessed using generalized linear models (GLM) followed by Tukey's honestly significant difference (HSD) test; Figure S3: Linear relationships between growing season (GS) length and end of studied tree groups separately and together; Table S1: Statistical significance presented as p-values derived from Tukey's honestly significant difference (HSD) test following the linear mixed effect models (LME) to assess difference in total year basal area increment (BAI), stem radius increment (GRO) and tree water deficit indicator (TWD) between individual tree groups. Significance also indicated as *** $p<0.001$, * $p<0.05$; Table S2: Output of the generalized mixed effect models (GLMM) of monthly data of growing rates (GRO) and tree water deficit indicator (TWD) as response variables and mean global radiation (GR), air temperature (T), vapor pressure deficit (VPD), precipitation totals (Prec) and water table depth (Depth) as fixed effects and years nested in trees as the random effects. Presented are estimates $\pm$ standard errors. Significance is indicated as *** $p < 0.001$, ** $p < 0.01$, * $p < 0.05$, '-' n.s.

**Author Contributions:** Conceptualization, R.G. and R.P.; Methodology, R.G. and R.P.; Formal Analysis, R.P.; Investigation, R.G. and R.P.; Resources, L.Ú. and R.P.; Data Curation, R.P.; Writing—Original Draft Preparation, R.P.; Writing—Review & Editing, R.G. and L.Ú.; Visualization, R.P.; Supervision, L.Ú. and R.G.; Project Administration, L.Ú.; Funding Acquisition, L.Ú. All authors have read and agreed to the published version of the manuscript.

**Funding:** This research was supported under the project "Significant Trees—Living Symbols of National and Cultural Identity" financed by the Ministry of Culture of the Czech Republic (project Reg. no. DG18P02OVV027).

**Data Availability Statement:** The data presented in this study are available on request from the corresponding author. The data are not publicly available due to obligations resulting from the project contract.

**Acknowledgments:** The authors are grateful to the Global Change Research Institute CAS for providing missing meteorological data, to the Czech Hydrometeorological Institute for providing the water table data and to Martin Šenfeldr and Martin Šrámek for the determination of tree age.

**Conflicts of Interest:** The authors declare no conflict of interest.

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
