# Peer review of "Stem Growth of Horse Chestnut (Aesculus hippocastanum L.) under a Warming Climate—Tree Age Matters"

_forests, doi:10.3390/f13101677_

Round 1
Reviewer 1 Report
GENERAL COMMENTS
This paper analyzed three different groups of horse chestnut daily stem growth from2019-2021 timespan and tried to explain the influence of the surrounding climate on radial increment performances. Firstly, this manuscript is within the scope of the Forests MDPI journal scope.
Drought-related studies are important for ongoing climate change. So, this is an important and hot topic for European foresters. Generally, I cannot see some difficulties with English. On the other hand, some structural things must be improved/added. This is mostly statistical difficulties with the aim to boost the reliability of the concluded statements.
In this step, I suggest to EiC a major revision, with the necessary second round of RW for this manuscript. Please, find my in detail RW below
SPECIFIC COMMENTS
Due to you analyzing trees in flooded areas, the river water level should be included in your research. I suggest adding this parameter to your manuscript.
Line (L) 104 I’m not sure that 50-year-old trees can be classified as young. Juvenile stage growth finished up to the 20th age. Should be reconsidered to find a better term for this group.
L 107-111 It is unclear why the authors analyzed only one younger-aged class and two different groups of older trees. It is very poorly described and must be improved. Also, if you analyzed border trees that can not be classified as a mature SOLITARY tree group.
Criteria for tree specimens’ selections are completely missing in sub-section 2.1.
L 113 Add abbreviations for mentioned parameters.
L 124 I wonder if is it correct to use and compare meteorological data from different sources. Please give an explanation or re-calculate some of the drought indices based on your measured data.
L 166-169 I cannot understand why you provide this information from your preliminary research. If you provide a reliable model outputs information of the used model superiority this other one is unnecessary only if you don’t both compare in the manuscript. Also, this part seems to me for the results section, not for M&M.
L 194 Why do you introduce a new (third) model here? I agree that non-linear models can be an appropriate choice for your data, but it is not explained as well. Also, if you decide to keep this model, add a smother explanation.
After reading the whole manuscript I didn’t find where you used explained models in sub-section 2.5. EXPLANATION
L 195-197 How do you calculate the ‘relationship’?
Fig 1. It seems to me that data of SPEI of two and/or three-month accumulated periods can be easier to explain drought trends. These is very drought-similar years to me.
Fig 2 change days to DOY on the y-axis.
L220-22-separated from the figure title.
Fig 3. This is completely wrong interpreted. This group cannot interpret together. What is the point to showed merged data?
Instead, this data (separated by groups) doesn’t have a linear trend. It is the easiest to see for the Mature-c group.
Sub-section 3.3. is the key section for this paper.
If authors want to compare differences among tree groups as well as years, statistics comparison is completely missed. I suggest to the authors prepare and add a new table with results of statistically significant differences among groups. E.g., Kendall tau correlation should be suitable for their data.
Fig. 4. Add what means a shaded area on the graph.
Sub-section 3.4. Some of the explained complex models in the M&M section should be a deeper explain the influence of climate parameters on steam growth than basic PCA.
This is too shortly explained and should be improved.
Although the conclusion section is not mandatory, I suggest authors add them at the end of the manuscript. It is too hard to understand what is concluding remarks of their research.
Author Response
We thank to reviewer for his valuable comments. Please, find attached detail responses.

Reviewer 2 Report
Dear authors,
Your manuscript makes a good impression of the extensive work completed, the topic is very important and actual. Below in pdf documents are some of my comments and suggestions.

Author Response

(The authors gave the same response as above.)

Round 2
Reviewer 1 Report
Dear authors,
The resubmitted manuscript is significantly improved. Mostly all difficulties were removed/changed, and this manuscript is close to the acceptable version. After re-reading the whole manuscript and the author’s reply to my RW I have some new suggestions and unclear methodological things.
The main new problem is a sampling strategy. As the authors add in the revised version in line 111, that they chose trees randomly is unacceptable to me. That means that they chose dominant and dominated trees, healthy and unhealthy.. etc. together into the same sample. For this type of study, it is unacceptable. A lot of paper was published about differences among the abovementioned tree groups. Again, for me, this is unacceptable.
I suggest EiC minor RW for this manuscript.
Author Response
REV1
The main new problem is a sampling strategy. As the authors add in the revised version in line 111, that they chose trees randomly is unacceptable to me. That means that they chose dominant and dominated trees, healthy and unhealthy.. etc. together into the same sample. For this type of study, it is unacceptable. A lot of paper was published about differences among the abovementioned tree groups. Again, for me, this is unacceptable.
We greatly appreciate this point of the reviewer. We agree that mentioned part of methodology is presented in unclear form and it was a big shortcoming. Of course, the trees were not selected fully randomly, but only trees representing the group were sampled. Healthy trees with the same social status and similar dimensions, and, regarding the canopy trees, dominant trees were sampled. From possible trees for study, the target ones were then selected randomly. We modified the corresponding text in manuscript as follow: “ … while another 5 dominant mature trees were chosen from the open canopy … The young canopy trees were not presented on our research area; thus, this group was omitted. Tree groups were selected to be as close as possible, to minimize the effect of the different site conditions, which are in this alluvial site minimal. Only representative healthy trees with the same social status and similar dimensions were selected within the studied groups.“